# CPMIP: Measurements of Real Computational Performance of Earth System Models in CMIP6

V. Balaji[1,4], E. Maisonnave[2], N. Zadeh[3,4], B. N. Lawrence[5,6], J. Biercamp[7], U. Fladrich[8], G. Aloisio[9,10], R. Benson[4], A. Caubel[11], J. Durachta[3,4], M.-A. Foujols[12], G. Lister[6], S. Mocavero[9], S. Underwood[3,4], and G. Wright[3,4]

[1]Princeton University, Cooperative Institute of Climate Science, Princeton NJ, USA
[2]Centre Européen de Recherche Avancée en Calcul Scientifique (CERFACS), Toulouse, France
[3]Engility Inc., Dover NJ, USA
[4]NOAA/Geophysical Fluid Dynamics Laboratory, Princeton NJ, USA
[5]National Centre for Atmopheric Science and University of Reading, UK
[6]Science and Technology Facilities Council, Abingdon, UK
[7]Deutsches Klimarechenzentrum GmbH, Hamburg, Germany
[8]Swedish Meteorological and Hydrological Institute, Norrköping, Sweden
[9]Centro Euro-Mediterraneo sui Cambiamenti Climatici (CMCC) Foundation, Lecce, Italy
[10]University of Salento, Lecce, Italy
[11]Laboratoire des Sciences du Climat et de l'Environnement LSCE/IPSL, CEA-CNRS-UVSQ, Université Paris-Saclay, 91191 Gif-sur-Yvette Cedex, France
[12]Institut Pierre-Simon Laplace, CNRS/UPMC, Paris, France

*Correspondence to:* V. Balaji (`balaji@princeton.edu`)

**Abstract.**

A climate model represents a multitude of processes on a variety of time and space scales; a canonical example of multi-physics multi-scale modeling. The underlying climate system is physically characterized by sensitive dependence on initial conditions, and natural stochastic variability, so very long integrations are needed to extract signals of climate change. Algorithms generally possess weak scaling and can be I/O and/or memory bound. Such weak-scaling, I/O and memory-bound multi-physics codes present particular challenges to computational performance.

Traditional metrics of computational efficiency such as performance counters and scaling curves do not tell us enough about real sustained performance from climate models on different machines. They also do not provide a satisfactory basis for comparative information across models.

We introduce a set of metrics that can be used for the study of computational performance of climate (and Earth System) models. These measures do not require specialized software or specific hardware counters, and should be accessible to anyone. They are independent of platform, and underlying parallel programming models. We show how these metrics can be used to measure actually attained performance of Earth system models on different machines, and identify the most fruitful areas of research and development for performance engineering.

We present results for these measures for a diverse suite of models from several modeling centers, and propose to use these measures as a basis for a CPMIP, a computational performance MIP.

# 1 Introduction

Climate and weather models (henceforth Earth System models or ESMs) have always been among the most computationally intensive scientific challenges. Strategic planning documents for high-performance computing such as André et al. (2014); Cappello et al. (2013); Attig et al. (2011); Reed and Dongarra (2015); Wehner et al. (2011) all outline the challenges presented by Earth system modeling to the coming generation of high-performance computing and data intensive computing.

ESMs are a computing and data challenge with a particular profile, as this article will show. The needs of ESMs are driven by trends in the science. Weather forecasting and the understanding of climate have both been synonymous with high-end computing since its pioneering days (Dahan-Dalmedico, 2001). Besides understanding the functioning of the Earth system, there are pressing needs on the science to serve other communities: ever since the seminal "Charney Report" of 1979 (Charney et al., 1979), the Earth system modeling community has also been increasingly responsive to the concerns about the human influence on climate. Computer simulations need to underpin scientific input to global policy decisions around possible mitigation and adaptation strategies. In the decades since, climate and weather have continued to be at the forefront of computational science, to be pioneering users of evolving supercomputing architectures, and drivers for data science.

As available computing power has continued to increase following "Moore's Law", so has the computing power demanded by Earth system modeling. ESMs consume computing along several axes, including *resolution*, as processes are included at finer and finer scale; *complexity* (to be defined more precisely below), as they seek to simulate, rather than prescribe, more and more processes and feedbacks internal to the climate system, and *ensemble size* to sample uncertainty across the chaotic non-linear dynamics that underlie complex systems. Where in this multi-dimensional domain of demand a given increase in computing is applied, depends both on the scientific problem of interest, but also, crucially, on the type of computer available. This is because different computing architectures are less or more suitable to increasing problem size along any of these axes. For example, a supercomputer with a fast communication fabric may be suitable for increasing resolution, as the fabric would support the increased communication load; whereas a more conventional loosely-coupled cluster may support a large ensemble of simulations that do not communicate amongst themselves. Whereas a novel machine with adequate fast memory may be able to accommodate the many variables and instructions associated with an increase in complexity.

Defining the mapping between the scientific problem and a computing architecture has become a crucial issue today. HPC architecture is at one of its transition points, or "disruptions". The previous transition, around two decades ago, moved HPC from the vector architectures of the Seymour Cray era, to distributed computing, based on networked clusters of commodity computers. The current transition is based on the end of how Moore's Law is traditionally understood (see e.g., Chien and Karamcheti, 2013), to a future where arithmetic and logic no longer gets faster on successive hardware generations, but may in fact get slower, alongside increases in parallelization and heterogenous memory architectures. On the current generation of new machines, ESMs have been able to show only modest gains in some measure of performance (Balaji, 2015). This means that traditional measures of computing power, such as flops (floating point operations per second) no longer appear to be representative of what is actually available.

In this article, we will examine the gaps between theoretical and actual performance (Section 2) and show how existing standard metrics of HPC performance are insufficient. We will demonstrate that there is sufficient diversity in ESMs so that no single measure, even a newly developed community one, is likely to be representative of the spectrum of ESMs. Rather we seek to identify a suite of measures for ESMs whose defining characteristics are:

- they are *universally* available from current ESMs, and applicable to any underlying numerics, as well as any underlying hardware architecture;

- they are representative of *actual performance* of the ESMs running as they would in a science setting, not under ideal conditions, or collected from representative subsets of code;

- they measure performance across the entire *lifecycle* of modeling, and cover both data and computational load;

- they are *easy to collect*, requiring no specialized instrumentation or software, but can be acquired in the course of routine production computing.

These measures are described in Section 3. In Section 4 we show results from many current ESMs. We conclude in Section 5 with a proposal to collect these metric routinely from the globally coordinated modeling campaigns such as the Coupled Model Intercomparison Project (CMIP: Meehl et al., 2000, now approaching its sixth generation in CMIP6). We hope thereby to outline a *computational and data profile for Earth System modeling* across the enterprise, which may be useful to define the kinds of machine most suited for this scientific and societal grand challenge in the exascale era.

## 2  Theoretical and actual computational performance

### 2.1  HPC performance measures: a brief history

The most common measure of computational performance is the theoretical maximum number of floating-point (FP) operations per second, or flops, achievable on a given machine. Computer vendors like to report this measure — peak flops — even though it is not achievable in practice. Peak flops are calculated by simply multiplying the number of arithmetic units (*arithmetic-logic units*, or ALUs) in hardware by the clock speed and any concurrency supported by the hardware (for example, *fused multiply-add* (FMA), or the *advanced vector extensions*, AVX, used in many modern processors to carry out multiple operations per clock cycle).

Unless the algorithm is perfectly tuned to the hardware layout, it is impossible to keep all ALUs and their internal hardware active all the time. A more practical measure is the maximum *sustained flops* that can be achieved with a real code. With the advent of parallel computing, the HPC community converged on a single code that was thought to be representative of compute intensive tasks, and compared between machines. This Linpack linear algebra benchmark (Dongarra, 1988) became the *de facto* HPC benchmark, and current supercomputer rankings, such as the Top 500 list are based on comparisons of measured sustained flops obtained running Linpack.

Very early in the parallel computing era it was recognized that even Linpack does not truly characterize real application performance (see e.g., the critique of the SPEC benchmarks in Dixit, 1991). One issue was the limitations imposed by memory bandwidth. Vector computers of the Seymour Cray era used specialized memory technology (called SRAM) to keep the vector registers filled. In the era of parallel computing, based on clusters constructed from commodity parts, it was often the case that bandwidth from commodity memory (DRAM) constrained computational performance more than computational speed itself. Accordingly, the STREAM benchmark (McCalpin, 1995) was developed to measure the performance obtained on FP codes when memory bandwidth is the limiting factor. This later led to a popular visual representation of performance limits imposed by both memory bandwidth and computational intensity known as the "roofline" (Williams et al., 2009).

Over time the community came to develop suites of kernels or "mini-apps" representing a spectrum of algorithms in use in HPC, such as the NAS Parallel Benchmarks (Bailey et al., 1991) and the HPC Challenge Suite (Luszczek et al., 2005). These were supposed to characterize a broad range of issues including clock speed, parallel arithmetic, memory bandwidth, cache efficiency, and the like. The kernel approach to getting a better measure of real computing performance has now converged on the HPCG benchmark (Dongarra et al., 2015), based on a popular elliptic solver, to supplement the HPC measure based on Linpack.

Despite all this progress, the key issue in measuring and improving computational performance remains the shortfall of actual performance obtained in real HPC applications relative to a theoretical ideal machine performance, often expressed as a *percent of peak*. The HPCG/HPC ratio, suitably normalized, is a good measure of this shortfall and has been steadily falling with each succeeding transition. While 50% of peak flops was attainable on Cray vector machines of the 1980s (and even NEC-SX machines into the current era), the figure of 10% was considered satisfactory in commodity parallel cluster architectures. The current transition toward fine-grained parallelism based on Graphical Processing Unit (GPU) and Many-Integrated Core (MIC) technology has pushed the "percent of peak" down into the single digits, as revealed by the HPCG/HPC ratio[1]. This trend warrants curbing one's enthusiasm when looking at peak-flops ratings of today's most powerful machines.

## 2.2 Computational performance of ESMs

Earth System Models have always presented a particular set of issues for performance on HPC architectures. To begin with, there is the problem of complexity. Climate science has been described as an attempt to simulate "the time evolution of the Earth system, a complex evolving mixture of fluids and chemicals in a very thin layer atop a wobbling, spinning sphere with an unstable surface and a molten interior, zooming through space in a field of extra-terrestrial photons at all wavelengths. Between sea and sky [lies] that thin layer of green scuzz that contain[s] all the known life in the universe, which itself [is] capable of affecting the state of the whole system." (Balaji, 2013) This growth in sophistication implies that the construction of an ESM (Figure 1) now involves large development teams, consisting of specialists in different aspects of the climate system such as atmospheric and oceanic dynamics, atmospheric chemistry, biosphere and land hydrology, and so on; with the whole system held together by a software framework. The framework may provide infrastructure services such as parallelism and I/O, as well as a *superstructure*, expressing the algorithms of coupling between components.

---

[1] "Architectural Surprises Underpin New HPC Benchmark Results.", HPCWire 2014-12-01

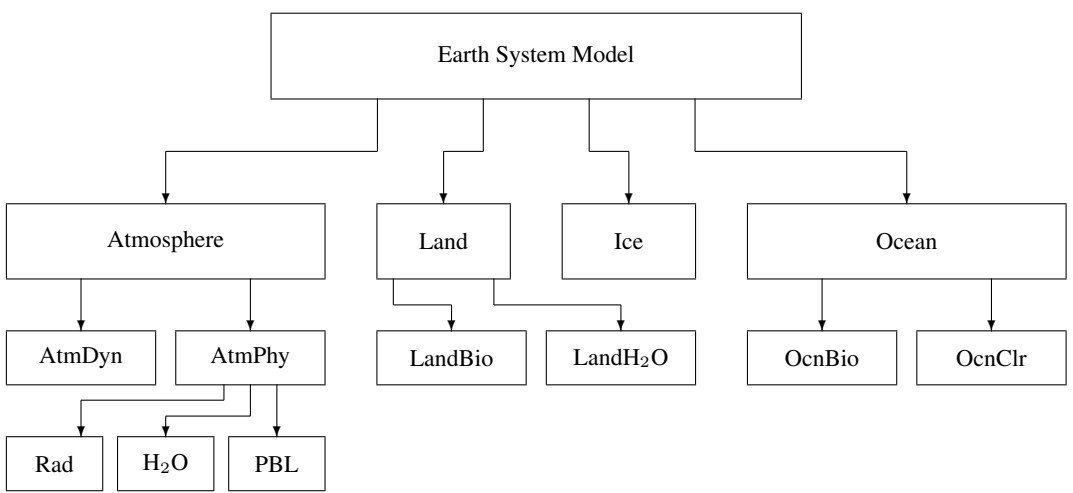

**Figure 1.** Notional architecture of an ESM: The model is composed of components (or sub-models) each of which is itself composed of components representing a group of one or more related processes. For example, within the atmosphere, the dynamical core (AtmDyn) is only one component, alongside the "physics" (AtmPhy), which itself has subcomponents for radiation (RAD), clouds and moisture ($H_2O$), planetary boundary layer (PBL), and so on. The list shown here is clearly not exhaustive, and one could easily imagine further recursive trees within any component shown.

The computational characteristics of ESM components can be quite diverse: a *land* component for instance may have no data dependencies across cells, but highly multivariate representations of ecosystem dynamics inside a cell; whereas an *atmospheric dynamical core* (*dycore*) may only encompass a few key variables representing momentum, mass and energy, but have strong cross-cell dependencies, which inhibit scaling. This is one reason why it is hard to define kernels, or "mini-apps", representative

5   of an ESM. Even for a single component, such as a dycore (which solves the equations of fluid flow for atmosphere or ocean), there is remarkable diversity of methods and approaches across models. Spectral, finite-difference (FD), finite-volume (FV), and finite-element (FE) methods are all in use in the world's major ESMs, as are both structured and unstructured grid approaches.

The dycore is quite often taken to be representative of the ESM as a whole. Dycores, regardless of numerics and mesh

10   choice, generally exhibit *weak scaling*, i.e., the concurrency achieved scales with the problem size[2]. Scaling is capped beyond some point for a fixed problem size, beyond which *strong scaling* is difficult to achieve. Thus, one may run a problem at higher *resolution* in the same time consuming more resources on a given machine (which might include a higher cost incurred in increased time resolution as well), but a model at fixed resolution is capped in terms of time to solution, absent advances in

---

[2]Since concurrency is usually achieved with a mixture of thread-based (shared-memory, such as OpenMP) and processor-based (distributed-memory, such as MPI) parallelism, we prefer to use the neutral term *concurrency* here, to indicate the number of concurrent executing elements, regardless of how this is achieved.

hardware or algorithm. Sometimes, adding more local physics (which does not involve distributed data dependencies) can improve scaling (the ability to use more computing capacity) but even that does not improve time to solution. We next address this issue of complexity of models as we go from dycores to full ESMs.

While dycores often consume the bulk of the resources devoted to performance engineering, their performance characteristics are not in fact representative of a whole ESM. This is because of the *complexity* inherent in climate modeling. Beyond the dycore, there are many other components. As shown in Figure 1, these may comprise the "physics", which is then further composed of components representing radiative transfer, clouds and connvection, and the planetary boundary layer (PBL), and so on. Many physical variables, of $\mathcal{O}(100)$ in modern ESMs, are needed to represent the full physics. Often these are local processes, which may not be a problem for scaling, but significantly alter the load per thread. Secondly, the number of variables (each typically a 3D array) is a significant burden on memory. The scaling behavior can be significantly different when "fully loaded" with physics.

A second feature of multi-component codes is that an ESM is quite often set up to run multiple component codes concurrently as separate excutables each with their own processor decomposition. This component architecture of ESMs is quite diverse (Alexander and Easterbrook, 2015), but typically most include at least two such components set up to run concurrently, in a mode we term *coarse-grained concurrency* (Balaji et al., 2016). This raises issues of *load balance*, configuring components to execute in roughly the same amount of time, so no processors sit idle. In such a "coupled" setting, components may not be able to run at their individual optimal scaling point, but rather at the scaling point which is optimal for the ESM as a whole. In addition, there are overheads associated with the coupling software itself. Components are generally allowed to have their own grid resolutions and timescales, and the coupler is responsible for exchanging information in a manner respecting numerical stability, accuracy, and above all, conservation of the quantities exchanged among the components. The coupling overhead must be taken into account in an ESM performance study. Coarse-grained concurrency may be increasingly prevalent in ESM architectures in the future, because of current hardware trends (Balaji et al., 2016). The parallel component layout of some typical ESMs is shown in Figure 2. Some components are scheduled to run concurrently, but usually are not exactly load-balanced, leaving load imbalance as blank spaces on the processor-time diagram, as shown. Other components may run serially at the end of other components. We will revisit this aspect of coupling below in Section 3.3.

A third consequence of complexity is that a large number of variables needs to be analyzed in scientific experiments involving ESMs. I/O is often ignored in scaling studies (including the standard HPC benchmarks other than those specifically measuring I/O performance), although rigorous and careful studies, such as the recent AVEC report, do take it into account. Both synchronous (blocking) and asynchronous (non-blocking) I/O subsystems are in use in ESMs. In the first instance, they will directly contribute to the measured time to solution, and in the latter, they will contribute to the cost in terms of additional processors devoted to I/O. In terms of real performance, it is important to include I/O, as the relative cost of computation and I/O is essential to defining a balanced machine suitable for ESMs.

We come to the third axis (see Section 1) along which Earth system modeling consumes computing power, that of *ensemble size*. The underlying dynamics of an ESM are *chaotic*, with a sensitive dependence on initial conditions, as has been known since the pioneering studies of Lorenz (1963). In climate and weather modeling, chaotic uncertainty is captured by running

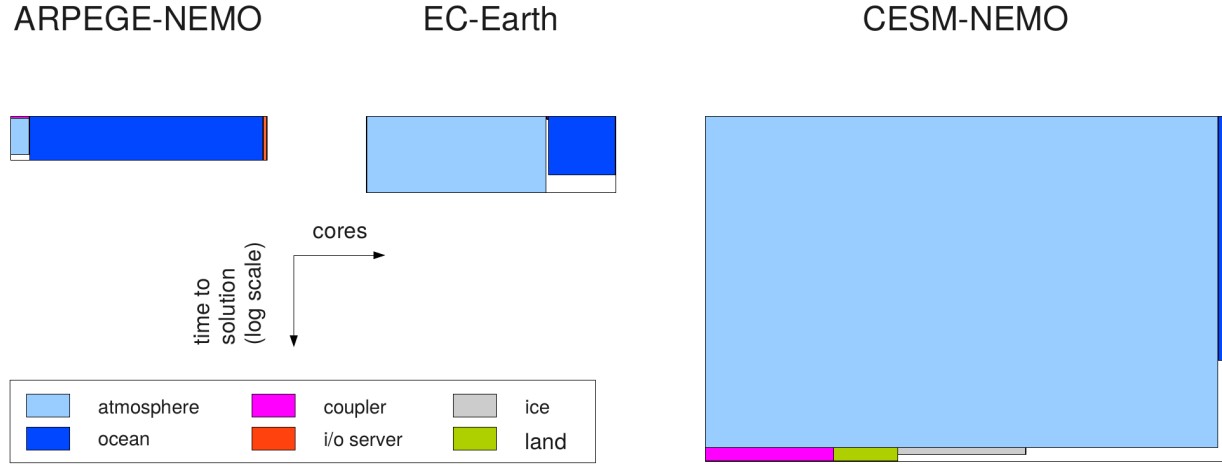

**Figure 2.** Component layout of three ESMs, in processor-time space (time increasing downward). Each box represents a component which is integrated either concurrently (coarse-grained concurrency, see text) in which case it is shown alongside the other components running at the same time, or sequentially, in which case it is shown below the previous component. The bounding rectangle shows the total cost of the coupled system, including waiting times due to load imbalance. Adapted from Fladrich and Maisonnave (2014).

an *ensemble* of simulations with slightly perturbed initial conditions, and examining the results in the form of a probability distribution, rather than an exact outcome. This has serious implications for understanding the performance of ESMs, as now the science is limited by the *capacity* of the computer (i.e., aggregated simulation time across an ensemble) rather than its *capability* (simulation time of a single instance). ESMs run for science may be run in both capability mode (fastest time to solution for a single instance) or capacity mode (best use of a computer allocation for an ensemble of runs), depending on need; both need to be assessed.

A final point regarding Earth system modeling is that runs may be resident on a system for very long times. Climate simulations often run for centuries or millennia of simulated time, taking wallclock time measured in months. This means that in actual practice, the time to solution is dependent on many factors, including the stability of the machine, the design of the queuing system, and the robustness of the workflow.

In summary, Earth system modeling has a particular computational and data *profile* which must be taken into account in measuring the computational "performance" of a given model on a given machine. The *profile* of climate computing is that of a *multi-scale, multi-physics* code, organized into a *hierarchy of components* that may be scheduled serially or concurrently, held together by sophisticated *coupling algorithms* that themselves carry a cost. Individual components generally exhibit *weak scaling*, are *memory-bound*, and may carry a significant *I/O load*. The models are executed for very long periods of time, so

that a significant cost is associated with the *workflow* and *machine policies* enabling sustained sequences of jobs. Finally, the models are sometimes run at their optimal speed, but quite often require large ensembles of simulations, so that they are in practice optimized for *capacity* rather than *capability*.

## 2.3 Real model performance: an alternate approach

The premise of this paper is that existing measures of computational performance do not give the Earth system science community adequate information about the actual model performance obtained in running production scientific runs. Such information is needed for a range of practical applications which go beyond the prediction of performance for traditional applications such as benchmarking new machines, to include the decisions needed to plan scientific experiments with real codes on specific hardware.

These features, required to assess real model performance in a scientific domain, require understanding the particular *domain computational profile* – in this case, the profile summarized at the end of Section 2.2 – and developing appropriate metrics to study performance.

Typical questions that ESM users have when they plan or run an experiment include:

– How long will the experiment take (including data transfer and post-processing)?

– How many nodes[3] can be efficiently used in different phases of the experiment?

– Are there bottlenecks in the experiment workflow, either from software or from system policies, such as queue structure and resource allocation?

– How much short-term/medium-term/long-term storage (disk, tape, etc.) is needed?

– Can/should the experiment be split up in parallel chunks (e.g., how many ensemble members should be run in parallel?)? What is the best use of my (limited) allocation?

Although these questions are clearly related to the computational performance of ESMs, they are not answered by examination of flops or speed-up curves.

We propose therefore an alternate approach. We have devised a set of computational performance metrics that directly address the concerns of this domain of science. The metrics have been chosen to satisfy several conditions:

– they are *universally* available from current ESMs, and applicable to any underlying numerics, as well as any underlying hardware architecture;

– they are representative of *actual performance* of the ESMs running as they would in a science setting, not under ideal conditions, or collected from representative subsets of code;

– they measure performance across the entire *lifecycle* of modeling, and cover both data and computational load;

---

[3]Although the number of computational elements is measured in *cores*, allocation is usually done in units of *nodes* of, say, 32 cores sharing memory.

– they are extremely *easy to collect*, requiring no specialized instrumentation or software, but can be acquired in the course of routine production computing.

These metrics will form the basis of a framework for routinely collecting these data from large coordinated modeling experiments. They are intended to serve as an adjunct to traditional, more idealized, measures of performance. They will allow the community as a whole to have a unified basis to evaluate technological advances through the lens of community concerns, and articulate community needs for computational and data architecture.

## 3   The CPMIP metrics

We propose below in Section 5 a systematic effort to collect metrics for a variety of climate models participating in common experiments. This proposed "model intercomparison project" (MIP) is to be called CPMIP: the Computational Performance MIP. The metrics proposed take into account the structure of ESMs and how they are run in production. Issues addressed include the following.

1. Models can have two optimal points of interest: one for speed (minimizing time to solution, maximizing simulated years per day or SYPD); the second for best use of a resource allocation (minimizing compute-hours per simulated year, or CHSY). A single ESM experiment may contain both phases. For instance, a climate experiment is often initialized from an idealized initial state, and a long "spinup" phase (measured in centuries for an AOGCM, millennia if the model includes a carbon cycle for instance, see e.g., Dunne et al., 2012) where we would run the model in *speed* or *capability* mode (the two terms are used interchangeably). After the spinup phase we have a near-equilibrium initial state of the climate which may be used to seed many experiments in parallel, in which case we would switch the configuration to *throughput* or *capacity* mode. We call these the S-mode and T-mode respectively.

   Figure 3 illustrates the S- and T-modes from a typical scaling study, in this case of a GFDL model configuration called `c96l32` running on a platform called `c3`. We see that the model is capable of running at 50 SYPD (simulated years per day, a quantity precisely defined below in Section 3.2). However by then, the scaling is beginning to suffer. In practice, we find that the best use of a computer allocation is to run the model at 35 SYPD, where the performance slope starts to change, indicating loss of scaling. The best throughput, measured in CHSY (also defined below in Section 3.2) is achieved at the lower processor count (1200 instead of 1600).

2. Computational cost scales with the number of *degrees of freedom* in the model. We factorize this number separately into *resolution* (number of spatial degrees of freedom) and *complexity* (number of prognostic variables). This separation is useful because performance varies inversely across resolution and complexity in weak-scaling models.

3. ESMs generally are configured to run more than one component concurrently: we need to measure load balance and coupler cost.

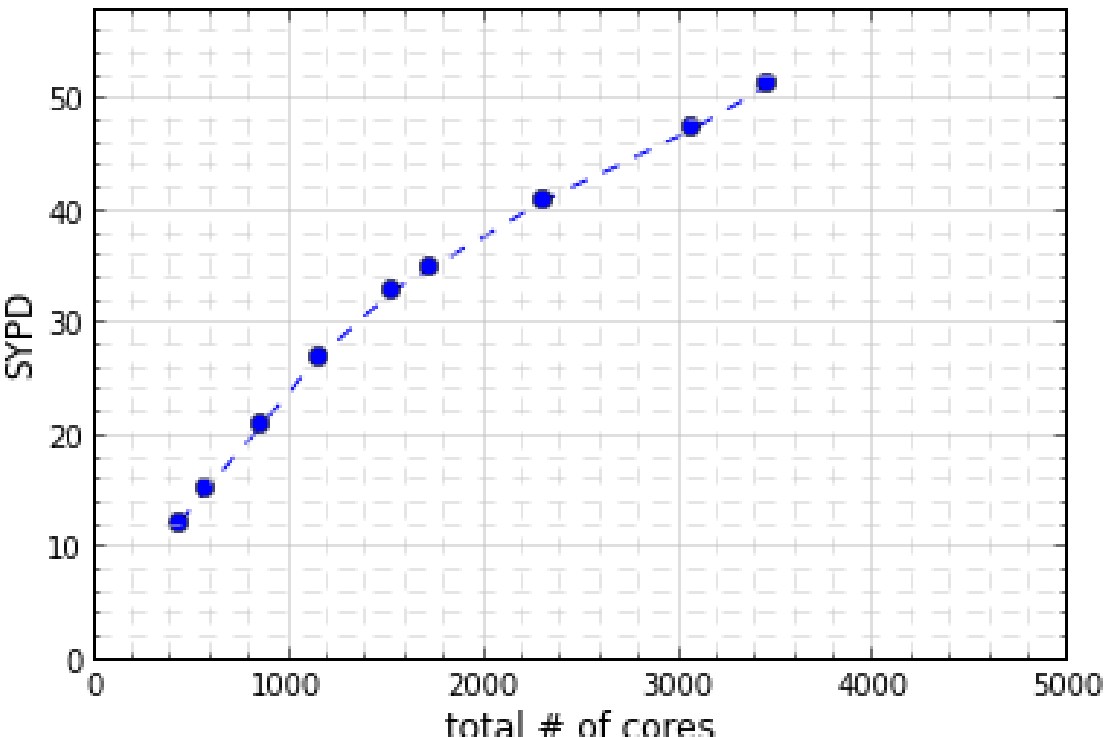

**Figure 3.** Scaling behaviour of a GFDL model. It illustrates that the model could be run at 50 SYPD in capability, or speed mode; but in practice is most often run at the shoulder of the curve, at around 35 SYPD, which gives the best throughput.

4. A vast number of variables is often used in the code, which is likely to aggravate the memory-boundness of models. While the theoretical minimum of one word (usually double-precision, or 8-byte) per variable per spatial degree of freedom is unavoidable, it is useful to measure memory *bloat*, excess copies of data made by the code, the compiler, or libraries. We do not necessarily consider the word *bloat* as pejorative: some of the extra copies might be needed for scientific reasons, such as halos (local caches of portions of neighboring domains in distributed memory), or providing registers for accumulating time-means of time-step data. But, reasons notwithstanding, these extra copies of data do indeed increase the memory requirements. And some data copies remain mostly outside user control (e.g., system I/O buffers).

5. Models configured for scientific analysis bear a significant I/O load, which can interfere with optimization of computational kernels. I/O may be synchronous (blocking) or asynchronous (non-blocking). Typical and maximum simulation data intensities (GB/CH) are useful measures for designing system architecture.

6. A full production climate model run, which may last weeks or months of investigator time, may be subject to delays not having to do with actual computational performance. We need a measure of these machine policy or workflow-related issues, a metric which indicates the need to devote resources to system (including management and queuing policies) and workflow issues rather than optimizing code.

To cover this list of concerns, we propose the following list of metrics, and indicate how this may be measured. (Recall that one prime consideration in the choice of metrics is the ease of collection).

## 3.1 The CPMIP Metrics: Model and Platform

We begin with metrics describing the model and the platform. The model is described by two basic characteristics:

**Resolution** measured as the number of gridpoints (or more generally, spatial degrees of freedom) $NX \times NY \times NZ$ per component, denoted by $G_c$ (where the subscript denotes a model component with an independent discretization). The resolution of the ESM is simply $G \equiv \sum_c G_c$.

$$
\begin{aligned}
G_c &\equiv NH \times NZ & (1) \\
G &\equiv \sum_c G_c & (2)
\end{aligned}
$$

where NH and NZ are the spatial degrees of freedom in the horizontal and vertical dimensions respectively. Given the small vertical extent of the atmosphere and ocean relative to the horizontal extent in any ESM, it is customary to represent them separately. NZ is thus the number of model levels; NH represents the horizontal degrees of freedom, which for instance may be $NX \times NY$ in a conventional FD or FV discretization, or the number of spherical harmonic elements retained under truncation in a spectral model; or the number of horizontal elements in an unstructured FE code.

We also additionally require a *representative* (noting that non-uniform grids are the norm) horizontal and vertical cell size for broad comparison purposes, $\Delta x_c$ and $\Delta z_c$, reported in km. The resolution is static information about the model and part of its configuration.

**Complexity** measured as the number of prognostic variables per component, $V_c$. This is also static, but if not available directly from the model configuration or code, it can be computed by dividing the size $S_c$ of the *restart file* (containing the complete state) per component, measured in words (e.g., 8 bytes for double precision) divided by $G_c$. The complexity of the model is $C \equiv \sum_c V_c$. The total degrees of freedom in the model is $F \equiv \sum_c G_c V_c$.

$$
\begin{aligned}
V_c &\equiv S_c / G_c / 8 & (3) \\
V &\equiv \sum_c V_c & (4)
\end{aligned}
$$

Note that the method of computing it from the restart file size assumes that only one copy of the model state is saved (i.e., no intermediate restarts). It further assumes that only one time-level of any variable is saved in the restart file. For models that use multiple time-level timestepping schemes, there could be several time levels saved in the restart file. For proper restarting of a model with leapfrog timestepping, for instance, both the current and prior state of a variable need to be saved. Thus, using the restart file method to estimate complexity is to be used with caution: it is better to have more direct methods of computing $V_c$, the number of prognostic variables.

Other methods for evaluating complexity (e.g., Méndez et al., 2014) are more based on evaluations of the model code itself, e.g., counting lines of source code. Our experience is that models of equal complexity in terms of the range of physical, chemical and biological process represented, vary considerably in terms of code, which appears to us not to provide a useful measure of complexity. We further note that most models are coded with a plethora of options, most of which are not exercised in any one model instance, thus resulting in a lot of "dead code".

The *platform* is a description of the computational hardware:

**Platform** There is a wide variety of machine descriptors which can be confusing. With the computing hierarchies in place, terms like *processor*, *processing element* or PE, and even computing *core*, become hard to compare across machines. However, the term core still has some universality as a concept, as a machine is often characterized by its *core count*, though what constitutes a core may not be strictly comparable across disparate hardware. With that concept, the two additional measures universally understood are the *clock speed* (usually reported in inverse time, so that larger is faster) in GHz, and the theoretically possible number of *double-precision* operations per clock cycle – which we term *clock-cycle concurrency*. All three measures can be obtained across the span of today's architectures, including GPUs, MICs, BlueGene, and conventional processors.

Note that there are additional numbers of interest, such as memory and filesystem characteristics. These are highly configuration-specific and quite often heterogeneous. We propose two additional descriptors: *chip name* (e.g., Knights Landing) and *machine name* (e.g., titan). These should allow one to find links to configuration-specific information about the platform.

## 3.2 The CPMIP Metrics: Computational Cost

**SYPD** simulated years per day for the ESM in a 24h period on a given platform. This should be collected by timing a segment of a production run (usually at least a month, often one or more years), not from short test runs. This is because short runs can give excessive weight to startup and shutdown costs, and distort the results following Amdahl's Law. This is measured separately in throughput and speed mode.

**ASYPD** the actual SYPD obtained from a typical long-running simulation with the model. This number may be lower than SYPD because of system interruptions, queue wait time, or issues with the model workflow. This is measured for a long

production run by measuring the time between first submission, and the date of arrival of the last history file on the storage filesystem. This is measured separately in throughput and speed mode. For a run of $N$ years in length

$$\text{ASYPD} \equiv \frac{N}{t_N - t_0} \tag{5}$$

where $t_0$ is the time of submission of the first job in the experiment, and $t_N$ is the timestamp of the history file for year $N$.

**CHSY** core-hours per simulated year. This is measured as the product of the model runtime for 1 SY, and the numbers of cores *allocated*.[4] This is measured separately in throughput and speed mode.

**Parallelization** measured as the total number of cores $NP$ allocated for the run. Note that $\text{NP} = \text{CHSY} * \text{SYPD}/24$.

**JPSY** is the energy cost of a simulation, measured in joules per simulated year. Energy is one of the key drivers of computing architecture design in the current era. While direct instrumentation of energy consumption on a chip is still something in development, we generally have access to the energy cost associated with a platform (including cooling, disks, and so on), measured in kWh (= $3.6 \times 10^6$ joules) over a month or a year. Given the energy $E$ in joules consumed over a budgeting interval $T$ (generally 1 month or 1 year, in units of hours), and the aggregate compute-hours $A$ on a system (total cores$*T$) over the same interval $T$, we can measure the cost associated with one year of a simulation as follows:

$$\text{JPSY} \equiv \text{CHSY} * E/A \tag{6}$$

Note that this is a very broad measure, and simply proportional to CHSY on a given machine. But it still is a basis of comparison *across* machines (as $E$ will vary). In future years as on-chip energy metering matures and is standardized, we can imagine adding an "actual joules per SY (AJPSY)" measure, which takes into account actual energy used by model and its workflow across the simulation lifecycle, including computation, data movement, and storage. These measures are similar in spirit to some prior measures of "energy to solution" (Bekas and Curioni, 2010; Cumming et al., 2014; Charles et al., 2015). The FTTSE metric of Bekas and Curioni (2010) is very similar to the AJPSY metric proposed here, and which we believe will replace JPSY in due course, when direct metering becomes routinely available.

### 3.3 The CPMIP Metrics: Coupling, memory and I/O

**Coupling cost** measures the overhead caused by coupling. This can include the cost of the coupling algorithm itself (which may involve grid interpolation and computation of transfer coefficients for conservative coupling) as well as load imbalance, when concurrent components finish at different rates, potentially leaving some PEs idle. It is possible to measure

---

[4]There may be some "rounding up" involved, as allocations usually are done on a node basis, and all cores on a node are charged against the allocation, regardless of whether or not they are used.

the two separately, but it involves somewhat subtle instrumentation, and may not be measurable in a uniform way across the range of ESM architectures used in the community (Alexander and Easterbrook, 2015). This is because load imbalance can manifest itself as time spent in the coupler (which is actually being spent in a spin-wait loop). We instead just choose to measure it as the normalized difference between the time-processor integral for the whole model versus the sum of individual *concurrent* components, or

$$C \equiv \frac{T_M P_M - \sum_c T_c P_c}{T_M P_M} \tag{7}$$

where $T_M$ and $P_M$ are the runtime and parallelization for the whole model, and $T_c$ and $P_c$ the same for individual components. Graphically, it can be seen as the "white area" for any ESM layout diagram like that of Figure 2. It involves a minimum of instrumentation to measure time spent in each component. These involve inserting simple timing calipers such as `MPI_WTime()` around components, excluding wait. While this may be considered extra "instrumentation", these are universally available basic routines, and indeed it would be a wonder if any ESM interested in performance did not have these embedded already.

**Memory Bloat** the ratio $B$ of the actual memory size to the ideal memory size $M_i$, defined below. The measured runtime memory usage $M$ on the system (often called "resident set size", or RSS) is divided between instructions and data, of which we are interested mainly in the latter. The RSS high-water mark is often published in the job epilog, failing which, we supply a small code (26 lines of C, see Section 6), which can be called at the end of a model run and will report the RSS high water mark on any linux-derived operating system. The portion of memory devoted to instructions is measured by taking the size of the executable files $X$ produced during compilation, of which one copy is stored on every processor. This may be an overestimate on systems where instructions are paged in, or shared between applications, so the correction for instruction size is to be applied with care. The ideal memory size is the size of the complete model state, which in theory is all you need to hold in memory, the rest being in principle computable from the state variables. Thus:

$$M_i \equiv \sum_c S_c \tag{8}$$

$$B \equiv \frac{M - \mathrm{NP} * X}{M_i} \tag{9}$$

Note that the "ideal" memory size is truly a utopian measure, and we never expect to get close in practice. Rather, it serves as a normalization factor allowing us to compare across different model characteristics (Section 3.1) and platforms. As we shall see, the value of $B$ is found to be $\mathcal{O}(10 - 100)$.

Unusually large numbers relative to other configurations can alert us to excessive buffering, and other issues. Also, we generally aspire to have *memory scaling* codes, where memory usage remains roughly constant across PE counts. It will generally not stay exactly constant because of the presence of halos. For instance, for a logically rectangular grid with a halo size of 2 in $X$ and $Y$, and a $20 \times 20$ domain under decomposition, the 2D array area including halos is 576 instead of 400, for a bloat factor of 1.44. The same model decomposed to a $10 \times 10$ domain, will have array area of 196 instead of 100, increasing the bloat to 1.96. This number might be somewhat larger for algorithms that use "wide halos" (Balaji, 2001). However, these factors are still small compared to the bloat caused by global arrays, which will cause memory to grow on a curve quadratic with the PE count (assuming 2D domain decomposition). Avoiding the use of global arrays is generally considered a useful approach in an era where memory movement is considerably more expensive than arithmetic.

**Data output cost** is the cost of performing I/O, and is the difference in cost between model runs with and without I/O. This is measured as the ratio of CHSY with and without I/O. This is measured differently for systems with synchronous and asynchronous I/O. For synchronous I/O where the computational PEs also perform I/O, it is requires a separate "No I/O" run, where we measure the fractional difference in cost:

$$D \equiv \frac{\mathrm{CHSY} - \mathrm{CHSY}_{\mathrm{noI/O}}}{\mathrm{CHSY}} \tag{10}$$

For models using asynchronous I/O such as XIOS, a separate bank of PEs is allotted for I/O. In this case, it may be possible to measure it by simply looking at the allocation fraction of the I/O server, without needing a second "no I/O" run.

$$D \equiv \frac{P_M - P_{\mathrm{I/O}}}{P_M} \tag{11}$$

However, there may be additional computations performed solely for diagnostic purpose; thus the method of Eq. 10 is likely more accurate. Note also that if the machine allocates by node, we need to account for the number of nodes, not PEs, allocated for I/O.

**Data intensity** the measure of data produced per compute-hour, in GB/CH. This is measured as the quotient of data produced per SY, easily obtained from examining the output directories, divided by CHSY.

# 4   Results from several ESMs

We present a spectrum of results from several ESMs to illustrate the power of the CPMIP approach. These are to be considered preliminary or suggestive findings.

Some of the metrics we collect are properties of the model which do not change however the model is run, but some are properties of the exact experiment for which it is used. In particular, the I/O properties (Data Output Cost and Data Intensity) will depend on the diagnostics required by the experiment.

Similarly, ASYPD is simulation dependent, depending not only on the model configuration but the background workload on the machine, which is the reason why we require this to be obtained from a long model run (so the background differences are averaged out).

For model intercomparison, the most understanding of model differences will be obtained when the differing models are being used for the same experiment. Hence, the full power of the method will only be apparent when we have systematically collected these metrics in conjunction with a major multi-center modeling project. A plan to do so in association with CMIP6 is outlined below in Section 5.

## 4.1 Speed and throughput modes

Performance results from HPC codes are often presented in the form of scaling curves, with time to solution plotted at various processor counts. A typical inference from such a plot is to identify models that scale well, i.e., close to an ideal scaling curve that points to the "strong scaling" limit. Recall that under strong scaling, the time to solution decreases inversely with the number of processors, i.e., half the time to solution for twice the assigned processing.

Most models scale less than perfectly, so in general scientific projects make compromises. There are two potential optima: one is to optimize time to solution by applying the maximum resource possible (the point at which the scaling curve saturates, so that adding more PEs does not improve time to solution); or alternately, pick a spot lower down the scaling curve for the maximum *aggregate* simulated years for an ensemble of model runs within a given allocation. In terms of the metrics defined in Section 3, we refer to these modes as the speed (S) or capability mode which maximizes SYPD, and the throughput (T) or capacity mode, which minimizes CHSY. Table 1 gives examples of the GFDL high resolution model CM2.6 (Griffies et al., 2015), for instance, which can be run at 2 SYPD, but in practice is most often run at 1 SYPD, which is the CHSY optimum. An ESM example is also shown (ESM2G, Dunne et al., 2012), where the 26 SYPD T configuration is usually run, but during model spinup (which is a single instance running, not an ensemble) the S configuration is used. ESM spinup often requires $\mathcal{O}(1000)$ years (Dunne et al., 2013), where raw speed is of the essence: we see that even at 40 SYPD a time on the order of months is needed simply to generate an equilibrated initial condition for a set of experiments.

## 4.2 Complexity, resolution and performance

We assume in the rest of the discussion that the runs being analyzed are in T-mode, as they would be run in production. In this section we show a comparison across several ESMs. The comparisons here necessarily have considerable scatter as they represent codes with differing levels of performance, and different hardware as well. Nonetheless, the inverse relationship between resolution and time to solution is seen in the scatter plot of Figure 4. Complexity, a second major determinant of performance, is shown as the size of the square on the scatter plot. Broadly, on similar performing hardware, we expect to see one group of models of limited complexity in one cluster on the resolution-SYPD slope, and another similar cluster for

| Model | Resol | Cmplx | SYPD | CHSY | Coupling | I/O | DI | MBloat | ASYPD | Platform |
|---|---|---|---|---|---|---|---|---|---|---|
| CM2.6 S | $4.9 \times 10^8$ | 18 | 2.2 | $2.12 \times 10^5$ | 26% | | 0.005 | | 1.6 | gaea/c2 |
| CM2.6 T | $4.9 \times 10^8$ | 18 | 1.1 | $1.81 \times 10^5$ | 62% | 24% | 0.005 | | 0.4 | gaea/c2 |
| CM2.5 T | $8.3 \times 10^7$ | 18 | 10.9 | 14327 | 17% | | | | 6.1 | gaea/c2 |
| FLOR T | $9.8 \times 10^6$ | 18 | 17.9 | 5844 | 57% | 5% | 0.015 | | 12.8 | gaea/c2 |
| CM3 T | $4.2 \times 10^7$ | 124 | 7.7 | 2974 | 42% | 15% | | | 4.9 | gaea/c2 |
| ESM2G S | $3.9 \times 10^6$ | 63 | 36.5 | 279 | 10% | 6.5% | 0.028 | 42 | 25.2 | gaea/c1 |
| ESM2G T | $3.9 \times 10^6$ | 63 | 26.4 | 235 | 25% | 6.5% | 0.028 | 42 | 11.4 | gaea/c1 |
| CM4H T | $1.2 \times 10^8$ | 57 | 6.9 | 7729 | 10% | 11% | 0.011 | 16 | 4.0 | gaea/c3 |
| CM4L T | $3.3 \times 10^7$ | 57 | 16.8 | 3277 | 20% | | 0.009 | 66 | 4.6 | gaea/c3 |
| ESM4L T | $3.3 \times 10^7$ | 104 | 10.1 | 5340 | 30% | | 0.013 | 40 | 7.7 | gaea/c3 |
| ARPEGE5-NEMO T | $1.2 \times 10^8$ | 18 | 5. | 5190 | 1% | 1% | 0.021 | 8.0 | 1.5 | curie |
| EC-Earth3.2 T | $1.4 \times 10^8$ | 34 | 1.3 | 12126 | 6.4% | 4% | 0.012 | 18 | 1.28 | beskow |
| EC-Earth3.2 S | $1.4 \times 10^8$ | 34 | 4.0 | 21481 | 11.0% | 1% | 0.007 | | 2.65 | beskow |
| CESM1.2.2-NEMO T | $1.2 \times 10^8$ | 103 | .86 | 59100 | 8.1% | 1% | $1.4 \times 10^{-3}$ | 9.1 | 0.04 | athena |
| MPI-ESM1 T | $2. \times 10^7$ | 73 | 18.5 | 3363 | 10% | 6% | 0.07 | 105 | 10 | mistral |
| NorESM1 S | $5. \times 10^6$ | | 17.2 | 1369 | | | | | | vilje |
| IPSL-CM6-LR S | $1. \times 10^7$ | 144 | 6 | 2166 | 5% | 10% | 0.01 | 9 | 5.5 | curie |
| HadGEM3-GC2 T | $1.8 \times 10^8$ | 66 | 1 | 6504 | 15% | | | | 0.57 | archer |

**Table 1.** Results from several ESMs. Not all the cells are currently filled, but we propose to collect these systematically in the full-scale CPMIP project. See text for explanation and discussion of terms.

high complexity models. Models lying considerably below the cluster representing their complexity class may indicate a need for performance improvement, either in the code or in hardware. In general, we can identify the low-complexity models as AOGCMs, and the high-complexity models as ESMs, which add chemistry and carbon to the mix. These results will of course be significantly clearer when we have a substantial database of results, allowing us to subselect based on platform, for example.

## 4.3 Energy consumption

We present here the JPSY metric for select models, with the caveats mentioned above in Section 3.2, namely that the energy costs are based on representative machine averages.

Table 2 shows the energy costs of the various model simulations in Table 1. The current results show that they are drawn from platforms with rather similar energetic profiles, and most of the variance in energetic costs come from variations in CHSY. Below in Section 4.7 we show a comparison of the same model on different platforms, with a substantial difference in energy profile. This will be seen to have significance in machine evaluation.

**Figure 4.** Performance, resolution and complexity for a subset of ESMs from Table 1, in throughput mode.

## 4.4 Coupler overhead and load imbalance

One area of concern in coupled modeling is the cost of the coupling itself. There are two aspects to this:

- When components are running concurrently there are synchronization costs which arise when the components must exchange data, i.e., a component that finishes early must wait for its boundary condition received from another component. Also components may have restrictions on the layout (i.e., the PE count can only be discretely altered). Second, the load is often a function of the actual narrative of events taking place in the model (e.g., convective activity). Thus it may not be possible to maintain an exact load match between components. This is usually done by trial and error and left fixed for the duration of an experiment.

- A second cost is that of the coupler itself: this includes the cost of conservative interpolation between independent model grids, as well as any other computations performed during the transfer. This can include computing fluxes, transforming quantities (for instance different components may have differing units or sign conventions for certain variables). In some cases transfer coefficients are computed on an intermediate "exchange grid" (e.g., Balaji et al., 2006).

| Model | CHSY | E | A | JPSY | Platform |
|---|---|---|---|---|---|
| CM2.6 S | $2.12 \times 10^5$ | $3.14 \times 10^{12}$ | $5.64 \times 10^7$ | $1.17 \times 10^{10}$ | gaea/c2 |
| CM2.6 T | $1.81 \times 10^5$ | $3.14 \times 10^{12}$ | $5.64 \times 10^7$ | $1.00 \times 10^{10}$ | gaea/c2 |
| CM2.5 T | 14327 | $3.14 \times 10^{12}$ | $5.64 \times 10^7$ | $7.99 \times 10^8$ | gaea/c2 |
| FLOR T | 5844 | $3.14 \times 10^{12}$ | $5.64 \times 10^7$ | $3.26 \times 10^8$ | gaea/c2 |
| CM3 T | 2974 | $3.14 \times 10^{12}$ | $5.64 \times 10^7$ | $1.66 \times 10^8$ | gaea/c2 |
| ESM2G S | 279 | $1.97 \times 10^{12}$ | $3.02 \times 10^7$ | $1.82 \times 10^7$ | gaea/c1 |
| ESM2G T | 235 | $1.97 \times 10^{12}$ | $3.02 \times 10^7$ | $1.53 \times 10^7$ | gaea/c1 |
| CM4H T | 7729 | $1.68 \times 10^{12}$ | $3.47 \times 10^7$ | $3.75 \times 10^8$ | gaea/c3 |
| CM4L T | 3277 | $1.68 \times 10^{12}$ | $3.47 \times 10^7$ | $1.59 \times 10^8$ | gaea/c3 |
| ESM4L T | 5340 | $1.68 \times 10^{12}$ | $3.47 \times 10^7$ | $2.59 \times 10^8$ | gaea/c3 |
| ARPEGE5-NEMO T | 5190 | $5.92 \times 10^{12}$ | $5.88 \times 10^7$ | $5.22 \times 10^8$ | curie |
| EC-Earth3.2 T | 12126 | $1.62 \times 10^{12}$ | $3.86 \times 10^7$ | $5.08 \times 10^8$ | beskow |
| EC-Earth3.2 S | 21481 | $1.62 \times 10^{12}$ | $3.86 \times 10^7$ | $8.99 \times 10^8$ | beskow |
| CESM1.2.2-NEMO T | 59100 | $9.00 \times 10^{12}$ | $6.76 \times 10^7$ | $7.87 \times 10^9$ | athena |
| MPI-ESM1 T | 3363 | $1.30 \times 10^{12}$ | $2.65 \times 10^7$ | $1.64 \times 10^8$ | mistral |
| NorESM1 S | 1369 | $1.41 \times 10^{12}$ | $1.64 \times 10^7$ | $1.18 \times 10^8$ | vilje |
| IPSL-CM6-LR S | 2166 | $5.92 \times 10^{12}$ | $5.88 \times 10^7$ | $2.18 \times 10^8$ | curie |
| HadGEM3-GC2 T | 6504 | $4.27 \times 10^{12}$ | $8.5 \times 10^7$ | $3.27 \times 10^8$ | archer |

**Table 2.** Energy cost per simulated year (joules) in several of the configurations listed in Table 1. See Section 4.3 for explanation of terms. Energy and Aggregate core-hours are reported for 1 month.

These are both unavoidable costs of coupling, therefore as outlined in Section 3 we have chosen to measure them as one: the coupling cost is the processor-time integral of the difference between the total cost of the coupled system, and the integrated cost of individual components, depicted graphically as the white area outside any component in Figure 2.

One area of concern is whether the coupler costs rise with resolution. A comparison of two models built from the same modeling system (the low-resolution ESM2G vs high-resolution CM2.6) in Table 1 show that the coupler cost including load imbalance, increases from 10% to 25% with the increase in resolution. This comparison is made in the S-mode. At lower PE counts (T-mode) it is more difficult to establish load balance because of layout restrictions as described above. Here the cost comparison across low and high resolution rises from 25% to 62%. Further examination indicates that this is an example of a model configuration that was insufficiently tuned for performance before starting a production run, and indeed a much better load balance could have been achieved. This inference is given a boost when we compare CM4H and CM4L, which are differentiated by high and low ocean resolution. Here in fact the coupling cost is *lower* in the high-resolution configuration. We therefore conclude that there is no evidence of a loss in coupling performance with resolution, and the anomalous result

for CM2.6 is probably due to an imperfect configuration. We present this as evidence that systematic collection of the CPMIP metrics would help identify such cases *during* setup for production, rather than *post facto*, as in this table.

## 4.5 I/O issues

As noted in Section 3, I/O load is measured here by comparing a production run with no diagnostic output against a regular production run. We see a generally modest cost ranging from 6.5% for low resolution models up to 24% at high resolution. (The CM2.6 run shown here contains an eddy-resolving ocean, and the high cost of I/O in that run is associated with high-frequency output for analyzing eddy statistics Griffies et al., 2015).

In other modeling systems with asynchronous I/O (such as the XIOS system developed in France, see Joussaume et al., 2012), the same cost is measured by seeing how many PEs are assigned to I/O relative to the rest of the model.

Another useful metric here is the data intensity defined in Section 3. It shows the rate of data production per hour of processing, in units of GB/CH. We see the data intensity decreasing as resolution increases, but staying proportional to increases in complexity.

## 4.6 Workflow costs

We see examples in Table 1 where there is a substantial discrepancy between ASYPD and SYPD, for instance the ESM2G T-mode  only achieves 11 SYPD in practice against an expected 26 SYPD. This indicates a need for closer examination. There could be several reasons for this:

- the workflow system could be introducing inefficiencies. This would be identified by a detailed examination of the run logs, to whether the delays are induced during data transfer or post processing, for instance.

- the queuing system could be introducing delays. Scheduler logs would identify whether there is excessive queue wait time, in which users may seek to change the queuing policies at their compute site, or else find a "sweet spot" for the T-mode  that best aligns with those policies.

- the run might have been interrupted by the scientist for various reasons, for example they might choose to "pause" the run to perform some preliminary analysis. In this case, we indeed discovered that there were significant gaps between output file timestamps at several points in the run, indicating that these were deliberate pauses.

These results indicate the utility of the CPMIP metrics for diagnosing problems associated with model workflow, that have as much impact on realized performance as algorithms and computational hardware.

## 4.7 Hardware comparison

One of the most impactful uses of the CPMIP metrics is in getting comparisons of actual performance improvements from new hardware. As we have emphasized in this paper, nominal measures of performance provided by vendors such as clock speed in GHz, or maximum theoretical flops, do not provide clear indications of what actual increase in  performance will be realized

| Model | Machine | Resol | SYPD | CHSY | JPSY |
|-------|---------|-------|------|------|------|
| CM4 S | gaea/c2 | $1.2 \times 10^8$ | 4.5 | 16000 | $8.92 \times 10^8$ |
| CM4 S | gaea/c3 | $1.2 \times 10^8$ | 10 | 7000 | $3.40 \times 10^8$ |
| CM4 T | gaea/c2 | $1.2 \times 10^8$ | 3.5 | 15000 | $8.36 \times 10^8$ |
| CM4 T | gaea/c3 | $1.2 \times 10^8$ | 7.5 | 7000 | $3.40 \times 10^8$ |

**Table 3.** Results comparing the same model in both speed (S) and throughput (T) mode on different hardware.

in practice on the actual applications run on the machine. In Table 3, we provide a direct comparison of the same codes on the current machine and on a new acquisition. NOAA has recently upgraded the technology on its flagship climate computer Gaea. The results of Table 1 were acquired on Gaea's `c1` and `c2` partitions in 2014, when it was a Cray XE6 (120,320 AMD Interlagos cores rated at 3.6 GHz, on a Cray Gemini fabric). In early 2016, a `c3` partition was added, a Cray XC40 consisting of 48,128 Intel Haswell cores rated at 2.3 GHz but with higher clock-cycle concurrency, and the next-generation Aries interconnect fabric (see Table 4). Given the higher rated processors and smaller number of cores, what is the true comparison across these machines?

Table 3 provides some answers. The CM4 model currently in development at GFDL shows a modest increase in cost as we increase the PE count, beginning to saturate in performance as we get to 4.5 SYPD (indicated by the increase in CHSY between rows 1 and 3). Over the same range, we are able to demonstrate an increase on the new hardware `c3` from 7.5 to 10 SYPD, at no increase in CHSY. We can infer three things:

– Core for core, the new machine shows a speedup of 2.2X, which one could not have inferred from the clock ratings. However the total number of cores has dropped 2.5X. Thus in aggregate, `c3` provides about 87% (2.2/2.5) of the capacity of the older `c1` and `c2` partitions combined, for the GFDL workload. Note, however, that the PF rating of `c3` (product of columns 3, 4, 5 in Table 4) is considerably higher than `c1` and `c2` combined (1.77 PF vs 1.12 PF). This shows the pitfalls of using petaflops ratings to infer aggregate performance of a machine.

– A second inference is that the next-generation network (Cray Aries over Gemini) is showing a manifest increase in performance, with the same CHSY in both configurations (i.e., with different numbers of PEs) whereas there was a drop in performance on the older hardware. Additional data on very high resolution models, not shown here, shows that the scaling increase results in vastly increased performance at very high PE counts, pushing the per-core performance difference to nearly 3X.

– A third and equally intriguing result is apparent from the energy analysis in the JPSY column. We see substantial decreases in the energy cost of simulation, with JPSY dropping by 60% in migrating from `c2` to `c3`, partly due to the lower CHSY, but also partly attributable to energy efficiency of the hardware. This translates into a very concrete and substantial fall in the total cost of simulation science over the lifetime of the machine.

| Machine | Chip | Cores | Clock speed (GHz) | Clock-cycle concurrency | URL |
|---------|------|-------|-------------------|-------------------------|-----|
| gaea/c1 | Interlagos | 41984 | 3.6 | 4 | `https://goo.gl/MYMPqD` |
| gaea/c2 | Interlagos | 78336 | 3.6 | 4 | `https://goo.gl/MYMPqD` |
| gaea/c3 | Haswell | 48128 | 2.3 | 16 | `https://goo.gl/o0xzIz` |
| curie | Sandy Bridge | 80640 | 2.7 | 8 | `http://goo.gl/RR5kfc` |
| mistral | Haswell | 36840 | 2.5 | 16 | `https://goo.gl/RKjC2g` |
| vilje | Sandy Bridge | 22464 | 2.6 | 8 | `https://goo.gl/ntxBPw` |
| athena | Sandy Bridge | 7712 | 2.6 | 8 | `https://goo.gl/Z2CSvB` |
| beskow | Haswell | 53632 | 2.3 | 16 | `https://goo.gl/ufDBBy` |
| archer | Ivy Bridge | 118080 | 2.7 | 8 | `http://goo.gl/dCU2uJ` |

**Table 4.** Details of platforms used in this study. The product of Cores, Clock Speed, and Clock-cycle concurrency should yield Theoretical Peak Speed, but as we see in the results of, and discussion around, tables Table 1 and Table 3, actual performance is rather different.

Comparisons of this nature based on CPMIP metrics can yield extremely useful material for comparing hardware and for interpreting benchmark results. A broad database of results across models and machines will also allow centers to gain useful insights about their own workload from acquisitions at other centers.

In future, modeling centers are likely to be distributing work across heterogeneous systems: this information could additionally aid in matching model configurations (e.g., resolution and complexity) to hardware.

The platforms used in this study are listed in Table 4.

## 5 Summary and future work

Computational performance is one of the most important constraints in the design of Earth system modeling experiments. These constraints force compromises between resolution, complexity and ensemble size, all of which have serious scientific implications. This paper proposes several metrics for assessing *real* computational performance of ESMs, and as an aid in experimental design and strategic planning, including future computer acquisitions consistent with a modeling center's mission.

It is our contention that traditional measures of computational performance do not provide the necessary input for experimental design and planning. The kinds of questions scientists face include:

– For a given experimental design, what can I afford to run?

– If I add complexity (such as adding a biogeochemistry component to an AOGCM), what will I have to sacrifice in resolution?

– How much computing capacity do I need to participate in a campaign like CMIP6 (Meehl et al., 2014)? How much data capacity?

– Do the queuing policies on the machine hinder the sustained run of a long-running model?

– During the spinup phase, how long (in wallclock time) before I have an equilibrium state?

The metrics we propose are designed to address questions such as these, not easily answered from flops and scaling curves. They are specifically designed to be universal (i.e., not based on a specific component hierarchy), very easy to collect (no specialized software or instrumentation), and reflective of actual performance in production.

As the energy cost of computing (Cumming et al., 2014; Charles et al., 2015) is increasingly becoming the limiting factor in large-scale computing, we expect that our machine-average measure of model energy consumption JPSY will need to be replaced by a more accurate measures of energy consumption AJPSY, using fine-grained hardware energy metering. In doing so, we will need to track energy consumption across the entire modeling lifecycle, including computation, data movement, and storage. Regardless of how energy per simulation unit is measured, we believe these metrics will play a substantial role in selecting technologies, as we will be able to demonstrate direct benefits in operating costs per "unit of science", such as a simulated year. Technologies that appear weaker in "core-for-core" comparisons may show up as stronger in energy comparisons. Across machines and models, we can imagine debates about certain choices of hardware being "slower but greener", or algorithms that are "less accurate but more eco-friendly". We believe these considerations will enrich the landscape of design of both hardware and simulation software and workflow.

Other questions may be asked as well, which are more project-specific. For instance, the CMIP experimental protocol is fundamentally dependent on an extensive process of data standardization, using tools such as the Climate Model Output Rewriter (CMOR). While the standardization is a tremendous boon to data consumers, the data producers often chafe at the somewhat onerous process of standardization. We could imagine project-specific metrics such as measuring the time spent making the CMIP runs, the total computational load of CMIP, and the time spent in post-model data standardization. The set of metrics may thus evolve in the future, with project-specific addenda.

We propose a systematic campaign to collect the basic metric set in this paper routinely for CMIP6 before considering its growth and evolution. This will be done using currently planned systems of model documentation such as ES-DOC (Lawrence et al., 2012). This comparative study of computational performance across models and machines, a CPMIP, will be an invaluable resource to the climate modeling community. Each center will individually be able to identify inefficiencies in their modeling lifecycle, and seek to address them. The comparative data will allow one center to predict the performance it will achieve on a machine available at another center. We propose to build such an emulator tool backed by the CPMIP database for this purpose. It will allow centers to define the optimal compute/data balance on future acquisitions. Finally, it will allow the Earth system modeling community as a whole to identify machine configurations and policies most apt to the kinds of science we hope to undertake in the future.

## 6 Data and code availability

The code for computing memory usage within the model (see Section 3.3) is provided in memuse.c. It is to be run on each PE and will provide the resident set size for that PE on any linux-based operating system.

All the data used in the tables and figures of this study are available in raw form in a public Dropbox folder. As stated in the paper, the ESDOC project will be collecting and publishing the data systematically during CMIP6 from all participating models.

*Acknowledgements.* V. Balaji is supported by the Cooperative Institute for Climate Science, Princeton University, Award NA08OAR4320752 from the National Oceanic and Atmospheric Administration, U.S. Department of Commerce. The statements, findings, conclusions, and recommendations are those of the authors and do not necessarily reflect the views of Princeton University, the National Oceanic and Atmospheric Administration, or the U.S. Department of Commerce. He is grateful to the Institut Pierre et Simon Laplace (LABEX-LIPSL) for support in 2015 during which drafts of this paper were written.

The research leading to these results has received funding from the European Union Seventh Framework program under the IS-ENES2 project (grant agreement No. 312979).

B.N. Lawrence acknowledges additional support from the UK Natural Environment Research Council.

We thank Lucas Harris and Ron Stouffer of NOAA/GFDL, as well as Claire Levy of IPSL and an anonymous reviewer, for insightful and thorough reviews of this article, down to the last comma.

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

   Good formats for the title include: "XYZMIP contribution to CMIP6: Name of project" or "Name of Project (XYZMIP) contribution to CMIP6"

   If you want to include a more descriptive title, the format could be along the lines of,

   "XYZMIP contribution to CMIP6: Name of project - descriptive title" or "Name of Project (XYZMIP) contribution to CMIP6: descriptive title."

   When you revise your manuscript, please add the reference to CMIP6 in the title of your manuscript accordingly.

   Additionally, we strongly recommend to add a version number to the MIP description. The reason for the version numbers is so that the MIP protocol can be updated later, normally in a second short paper outlining the changes. See, for example: http://www.geosci-model-dev.net/special_issue11.html

   See page 1, line 1. We have modified the title as requested. We envision the CPMIP database to accumulate results across model generations, so we we could like to keep the same name even if the version changes.

Response to gmd-2016-197-RC1:

2. Page 2 line 20: "This is because different computing architectures are less or more suitable to increasing problem size along any of these axes." My reading did not allow me to understand this sentence concluding the paragraph. Maybe a little more explanation would help (me, at least)

See page 2, line 20. We agree that this line is a bit obscure, and have added a paragraph of explanation and examples.

3. Page 3 line 26: "One issue was the limitations imposed by memory." I suggest replacing memory by memory bandwith (e.g. not memory size)

Agreed, see page 4, line 3.

4. Page 4 and 5: I'm not fully convinced by the Figure 1: is seems incomplete (and difficult to complete!). One could add a number of boxes (OcnDyn, OcnPhy, IceDyn, IceBio, etc...) so as a number of couplings, interactions and related processes. Would an "incomplete list" be better, or is there a better way to build a picture to make this complex ESM architecture visible?

We have added additional text to the caption to Fig. 1 to make it clear that the tree shown in this diagram could have further embedded components.

5. Page 5 line 8: "but a model at fixed resolution is capped in terms of time to solution, absent advances in hardware or algorithm." I agree if you are still talking about one dycore in this paragraph, but this is not true for a whole ESM.

Our understanding of this comment is that adding complexity can improve scaling, but in fact it cannot improve time to solution. We have added some text to explain this, and connect to the next paragraph, which goes in depth into the issue of model complexity. See page 6, line 1.

6. Page 6 Figure 2: Not clear to me what is the conclusion of paragraph ending line 14, nor precisely what is demonstrated on Figure 2.

This is a fair comment, as the discussion around Fig. 2 foreshadows a discussion to come, further down in Sec. 3.3. We have added some lines to make the link to the discussion of coupling cost more explicit, which we hope addresses the concern raised here. See page 6, line 23. We have also modified Fig. 2 to show the bounding rectangle.

7. Page 8: I believe the answers to the questions listed at top of page depend on how the computer platform is used: in dedicated mode, or not? But this seems to be taken in account in the ASYPD metric?

Yes, this is correct. The ASYPD metric is supposed to reveal performance issues associated with such "policy" issues such as whether the target machine is in "dedicated mode" as the reviewer has stated. In particular, we wish to see ASYPD measured just as it is run in production, not under ideal conditions: so one should not take the measurement in dedicated mode, if in practice the machine is not dedicated! See also page 8, line 1; page 11, line 4; page 13, line 2; page 16, line 6; page 20, line 21; page 23, line 1.

8. Page 11: taking the resolution in account Here, you have only taken the spatial resolution in account. I wonder if somehow (not simple though), temporal resolution should be take in account in the G metric: using a large or a small time step does indeed widely change the number of floating point operation you need for one simulated year. Now, what time step to use (those are different in atmosphere, ocean, coupler...)? I do not have any simple answer, but to be able to compare metrics between ESMS, I suggest to add something in G to take in account number of time steps per year as an unavoidable constraint on the number of floating point operations. (Could also be in a next generation of those metrics)

We do not believe the temporal resolution should be included in $G$, for the following reason: it is chosen as a consequence of the spatial resolution and the accuracy one needs to achieve. For example, the timestep may be bounded by a CFL criterion once the spatial resolution is given, for one class of methods. For implicit methods where there is no CFL, it is still bounded by the level of accuracy needed. In any case, the timestep is never varied independently of the limits of physics and resolution purely for reasons of speed and performance, and is generally set to the largest value imposed by those limits. It therefore should not be separately accounted for in $G$.

9. Page 19: Reference for XIOS: is this paper from Joussaume et al. the appropriate reference for XIOS today?

Unfortunately, there does not yet exist a better reference to XIOS than Joussaume et al. (2012) and the URL to the source and documentation, already provided.

Response to gmd-2016-197-RC2: (note that some internal numbering from the review has been modified to create consistent numbering.

10. One can still debate whether other metrics should be considered as relevant to our field. While the paper correctly suggests the increasing importance of energy metrics in HPC – in this case the JPSY and AJPSY metrics – perhaps it should also consider other evolving metrics, e.g., FTTSE [Bekas, C. & Curioni, A. Comput Sci Res Dev (2010) 25: 187]. This comment is meant to encourage discussion and promote the evolution of a more complete set of metrics. As mentioned, the strength of the paper lies in the agreement of many institutions on the presented set.

    This is indeed a useful reference, and has been added to the text. As noted in the discussion of AJPSY (which is very similar to the FTTSE), we believe we will extend toward such metrics when the hardware routinely supports it. See revised discussion starting page 13, line 20.

11. Explicit mention of spectral models should be made on p. 5, line 1. Spectral models are still in use and tend to have very different scaling behaviour than FD-, FV- or FEbased models.

    Agreed, this is an oversight. Fixed, see page 5, line 6.

12. Point 6 on p. 10 (lines 4-6) is not necessary to mention here. It is repeated in section 3.2 and need not be emphasised.

    Agreed, we have reworded this section to make the general point instead of repeating what is below in Sec 3.2. See page 11, line 3.

13. Given the caveats mentioned about dimensionality on p. 11, lines 6-7, why even mention the nominal Cartesian representation of NX x NY x NZ on p. 11, line 1?

    Fair point: see paragraph of revised text beginning page 11, line 14.

14. There are several full sentences enclosed in parentheses (p. 12, lines 28-29; p. 14, lines 7-8; p. 15, lines 25-26; p. 19, lines 19-21). As a matter of style, it would be better to either (a) remove the parentheses if the authors feel these comments belong in the text, or (b) make them footnotes.

    Fixed; the parentheses have been replaced by a footnote in one instance (page 13, line 7); sentence eliminated altogether just below, as it repeats the footnote text (page 13, line 8); parentheses removed and sentence retained in two instances (page 14, line 19; page 16, line 14).

15. The paper is well-written and concise. Obviously great care has been taken in proofreading. Still a small number of mistakes have slipped through. GMD allows authors to write in the variety of English of their choice: We accept all standard varieties of English in order to retain the author's voice. However, the variety should be consistent within each article. When using Oxford spellings, please do so consistently. Whilst the vast majority of spellings are U.S. ("parallelization", "modeling", "characterize", "recognize", ...), the authors curiously use "centre" (8 occurrences, although 2 occurrences of "center"), and "summarised" (1 occurrence). Please use consistent American spellings, if this is your choice. Fixed;

see page 8, line 11; page 1, line 16; page 22, line 4; page 22, line 12; page 23, line 25; page 23, line 26; page 23, line 27; page 23, line 28. Additionally we found one instance of "modelling" which has now become "modeling", consistent with the rest of the text: see page 2, line 10.

16. Various forms are used for "flops", such as "FLOP rates", "flop rates", and "flops". Please use one consistently.

    We have now used "flops" consistently: see page 8, line 22; page 20, line 30; page 23, line 4.

17. While there is no consensus in writing manuals whether there should be a comma after "i.e." and "e.g.", it is universally agreed that there should be a full stop after each letter. Please correct the instances of "i.e" and "e.g".

    Fixed, see page 2, line 29; page 4, line 2; page 5, line 10; page 7, line 3; page 8, line 19; page 9, line 15; page 10, line 7; page 11, line 23; page 12, line 7; page 12, line 8; page 12, line 22; page 12, line 23; page 16, line 14; page 16, line 15; page 18, line 4; page 18, line 5; page 18, line 6; page 18, line 12; page 21, line 18; page 22, line 5; page 23, line 4.

18. The four points at the end of p. 2 / beginning of p. 3 are part of the sentence on p. 2, lines 31-32. Please terminate these with semi-colons rather than full stops.

    Fixed; see page 3, line 6; page 3, line 8; page 3, line 9.

19. Same point for the 4 points in the sentence on p. 8, lines 15-22.

    Fixed; see page 8, line 26; page 8, line 28; page 8, line 29.

20. p. 7, line 1: "such as the recent AVEC report do take it " should be "... recent AVEC report, do take it".

    Fixed; see page 6, line 29.

21. p. 7, line 12: "... maybe run in ..." should be " ... may be run in ... ".

    Fixed; see page 7, line 5.

22. p. 15, line 1: "In this case. it may be ..." should be "In this case, it may be ...".

    Fixed; see page 15, line 17.

23. "S-mode" and "T-mode" are defined (quite appropriately!) on p. 9, line 13, but are used as "S mode" (p. 19, line 8) and "T mode" (p. 16, line 10; p. 19, lines 28-29). Please use consistently.

    Fixed; see page 16, line 28; page 19, line 6; page 20, line 15; page 20, line 21.

24. p. 21, line 1: "core for core, the new machine ..." should be "Core for core, the new machine ...".

    Fixed; see page 21, line 12.