# Peer review of "CPMIP: Measurements of Real Computational Performance of Earth System Models in CMIP6"

_Geoscientific Model Development, 2016_

## Short Comment (SC1) · 13 Sep 2016

Dear authors,

In agreement with the CMIP6 panel members, the Executive editors of GMD would like to establish a common naming convention for the titles of the CMIP6 experiment description papers.

The title of CMIP6 papers should include both the acronym of the MIP, and CMIP6, so that it is clear this is a CMIP6-Endorsed MIP.

Good formats for the title include:

'XYZMIP contribution to CMIP6: Name of project'

or

'Name of Project (XYZMIP) contribution to CMIP6'

If you want to include a more descriptive title, the format could be along the lines of,

'XYZMIP contribution to CMIP6: Name of project - descriptive title'

or

'Name of Project (XYZMIP) contribution to CMIP6: descriptive title.'

When you revise your manuscript, please add the reference to CMIP6 in the title of your manuscript accordingly.

Additionally, we strongly recommend to add a version number to the MIP description. The reason for the version numbers is so that the MIP protocol can be updated later, normally in a second short paper outlining the changes. See, for example: http://www.geosci-model-dev.net/special_issue11.html,

Yours,

Astrid Kerkweg

---

## Referee Comment (RC1) · C. Levy (Referee) · 5 Oct 2016

General comments I believe this paper is a major breakthrough for ESMs and climate studies: it introduces a set of metrics universally available for ESMs in the play for CMIP6. Using this suggestions for CMIP6 will be of great help for the future. All the required expertise is shown, allowing the authors to propose a set of very relevant metrics. The paper is nicely written and easy to read. It should be published and publicised in time (= as soon as possible now) to allow the whole community to set up the proposed metrics for the CMIP6 core experiments. It could be followed in the next years by some updates on those useful metrics (from adding something on temporal resolution (see below), to emerging ideas needing further discussions (single precision...)

Specific comments:

Page 2 line 20: "This is because different computing architectures are less or more suitable to increasing problem size along any of these axes." My reading did not allow me to understand this sentence concluding the paragraph. Maybe a little more explanation would help (me, at least)

Page 3 line 26: "One issue was the limitations imposed by memory." I suggest replacing memory by memory bandwith (e.g. not memory size)

Page 4 and 5: I'm not fully convinced by the Figure 1: is seems incomplete (and difficult to complete!). One could add a number of boxes (OcnDyn, OcnPhy, IceDyn, IceBio, etc...) so as a number of couplings, interactions and related processes. Would an "incomplete list" be better, or is there a better way to build a picture to make this complex ESM architecture visible?

Page 5 line 8: "but a model at fixed resolution is capped in terms of time to solution, absent advances in hardware or algorithm." I agree if you are still talking about one dycore in this paragraph, but this is not true for a whole ESM.

Page 6 Figure 2: Not clear to me what is the conclusion of paragraph ending line 14, nor precisely what is demonstrated on Figure 2.

Page 8: I believe the answers to the questions listed at top of page depend on how the computer platform is used: in dedicated mode, or not? But this seems to be taken in account in the ASYPD metric?

Page 11: taking the resolution in account Here, you have only taken the spatial resolution in account. I wonder if somehow (not simple though), temporal resolution should be take in account in the G metric: using a large or a small time step does indeed widely change the number of floating point operation you need for one simulated year. Now, what time step to use (those are different in atmosphere, ocean, coupler...)? I do not have any simple answer, but to be able to compare metrics between ESMS, I suggest to add something in G to take in account number of time steps per year as an

unavoidable constraint on the number of floating point operations. (Could also be in a next generation of those metrics)

Page 19: Reference for XIOS: is this paper from Joussaume et al. the appropriate reference for XIOS today?

---

## Referee Comment (RC2) · Anonymous Referee #2 · 31 Oct 2016

In any scientific field, it is crucial to step back periodically from research to find community agreement on how to evaluate results. This is no different in the field of modelling: The hard work of numerous research groups has given us a plethora of earth system models, varying greatly in their underlying algorithms and their approach to exploiting parallel architectures. Yet thought was rarely given on how their performance can or should be compared.

This paper represents such a step back: It proposes a set of metrics which can be used to study the computational performance of models without introducing dependencies on computer platforms or programming paradigms. The importance of the contribution lies not in the metrics themselves, for practitioners in the field will recognise most of them for they are already (e.g., SYPD, ASYPD and CHSY) used in daily modelling discourse. It lies rather in the consensus of the authorship from numerous international institutions, which implies an agreement on the greatest common denominator. The metrics are common-sense, and their calculation and application has been documented in a straigtforward manner. Moreover, the metrics are relevant in the framework of CMPI6. For all of these reasons the paper should be published at the earliest possible convenience.

One can still debate whether other metrics should be considered as relevant to our field. While the paper correctly suggests the increasing importance of energy metrics in HPC – in this case the JPSY and AJPSY metrics – perhaps it should also consider other evolving metrics, e.g., FTTSE [Bekas, C. & Curioni, A. Comput Sci Res Dev (2010) 25: 187]. This comment is meant to encourage discussion and promote the evolution of a more complete set of metrics. As mentioned, the strength of the paper lies in the agreement of many institutions on the presented set.

A few structural improvements could be made as part of a minor revision:

1) Explicit mention of spectral models should be made on p. 5, line 1. Spectral models are still in use and tend to have very different scaling behaviour than FD-, FV- or FE-based models.

2) Point 6 on p. 10 (lines 4-6) is not necessary to mention here. It is repeated in section 3.2 and need not be emphasised.

3) Given the caveats mentioned about dimensionality on p. 11, lines 6-7, why even mention the nominal Cartesian representation of NX x NY x NZ on p. 11, line 1?

4) There are several full sentences enclosed in parentheses (p. 12, lines 28-29; p. 14, lines 7-8; p. 15, lines 25-26; p. 19, lines 19-21). As a matter of style, it would be better to either (a) remove the parentheses if the authors feel these comments belong in the text, or (b) make them footnotes.

The paper is well-written and concise. Obviously great care has been taken in proof-reading. Still a small number of mistakes have slipped through.

1) GMD allows authors to write in the variety of English of their choice:

We accept all standard varieties of English in order to retain the author's voice. However, the variety should be consistent within e ach article. When using Oxford spellings, please do so consistently. Whilst the vast majority of spellings are U.S. ("parallelization", "modeling", "characterize", "recognize", ...), the authors curiously use "centre" (8 occurrences, although 2 occurrences of "center"), and "summarised" (1 occurrence). Please use consistent American spellings, if this is your choice. 2) Various forms are used for "flops", such as "FLOP rates", "flop rates", and "flops". Please use one consistently.

3) While there is no consensus in writing manuals whether there should be a comma after "i.e." and "e.g.", it is universally agreed that there should be a full stop after each letter. Please correct the instances of "i.e" and "e.g".

4) The four points at the end of p. 2 / beginning of p. 3 are part of the sentence on p. 2, lines 31-32. Please terminate these with semi-colons rather than full stops.

5) Same point for the 4 points in the sentence on p. 8, lines 15-22.

6) p. 7, line 1: "such as the recent AVEC report do take it " should be "... recent AVEC report, do take it".

7) p. 7, line 12: "... maybe run in ..." should be " ... may be run in ... ".

8) p. 15, line 1: "In this case. it may be ..." should be "In this case, it may be ...".

9) "S-mode" and "T-mode" are defined (quite appropriately!) on p. 9, line 13, but are used as "S mode" (p. 19, line 8) and "T mode" (p. 16, line 10; p. 19, lines 28-29). Please use consistently.

10) p. 21, line 1: "core for core, the new machine ..." should be "Core for core, the new machine ...".

---

## Author Comment (AC1) · 26 Nov 2016

Response to gmd-2016-197-SC1:

1. In agreement with the CMIP6 panel members, the Executive editors of GMD would like to establish a common naming convention for the titles of the CMIP6 experiment description papers. The title of CMIP6 papers should include both the acronym of the MIP, and CMIP6, so that it is clear this is a CMIP6-Endorsed MIP.

   Good formats for the title include: "XYZMIP contribution to CMIP6: Name of project" or "Name of Project (XYZMIP) contribution to CMIP6"

   If you want to include a more descriptive title, the format could be along the lines of,

   "XYZMIP contribution to CMIP6: Name of project - descriptive title" or "Name of Project (XYZMIP) contribution to CMIP6: descriptive title."

   When you revise your manuscript, please add the reference to CMIP6 in the title of your manuscript accordingly.

   Additionally, we strongly recommend to add a version number to the MIP description. The reason for the version numbers is so that the MIP protocol can be updated later, normally in a second short paper outlining the changes. See, for example: http://www.geosci-model-dev.net/special_issue11.html

   See page 1, line 1. We have modified the title as requested. We envision the CPMIP database to accumulate results across model generations, so we we could like to keep the same name even if the version changes.

---

## Author Comment (AC2) · 26 Nov 2016

Response to gmd-2016-197-RC1:

2. Page 2 line 20: "This is because different computing architectures are less or more suitable to increasing problem size along any of these axes." My reading did not allow me to understand this sentence concluding the paragraph. Maybe a little more explanation would help (me, at least)

See page 2, line 20. We agree that this line is a bit obscure, and have added a paragraph of explanation and examples.

3. Page 3 line 26: "One issue was the limitations imposed by memory." I suggest replacing memory by memory bandwith (e.g. not memory size)

Agreed, see page 4, line 3.

4. Page 4 and 5: I'm not fully convinced by the Figure 1: is seems incomplete (and difficult to complete!). One could add a number of boxes (OcnDyn, OcnPhy, IceDyn, IceBio, etc...) so as a number of couplings, interactions and related processes. Would an "incomplete list" be better, or is there a better way to build a picture to make this complex ESM architecture visible?

We have added additional text to the caption to Fig. 1 to make it clear that the tree shown in this diagram could have further embedded components.

5. Page 5 line 8: "but a model at fixed resolution is capped in terms of time to solution, absent advances in hardware or algorithm." I agree if you are still talking about one dycore in this paragraph, but this is not true for a whole ESM.

Our understanding of this comment is that adding complexity can improve scaling, but in fact it cannot improve time to solution. We have added some text to explain this, and connect to the next paragraph, which goes in depth into the issue of model complexity. See page 6, line 1.

6. Page 6 Figure 2: Not clear to me what is the conclusion of paragraph ending line 14, nor precisely what is demonstrated on Figure 2.

This is a fair comment, as the discussion around Fig. 2 foreshadows a discussion to come, further down in Sec. 3.3. We have added some lines to make the link to the discussion of coupling cost more explicit, which we hope addresses the concern raised here. See page 6, line 23. We have also modified Fig. 2 to show the bounding rectangle.

7. Page 8: I believe the answers to the questions listed at top of page depend on how the computer platform is used: in dedicated mode, or not? But this seems to be taken in account in the ASYPD metric?

Yes, this is correct. The ASYPD metric is supposed to reveal performance issues associated with such "policy" issues such as whether the target machine is in "dedicated mode" as the reviewer has stated. In particular, we wish to see ASYPD measured just as it is run in production, not under ideal conditions: so one should not take the measurement in dedicated mode, if in practice the machine is not dedicated! See also page 8, line 1; page 11, line 4; page 13, line 2; page 16, line 6; page 20, line 21; page 23, line 1.

8. Page 11: taking the resolution in account Here, you have only taken the spatial resolution in account. I wonder if somehow (not simple though), temporal resolution should be take in account in the G metric: using a large or a small time step does indeed widely change the number of floating point operation you need for one simulated year. Now, what time step to use (those are different in atmosphere, ocean, coupler...)? I do not have any simple answer, but to be able to compare metrics between ESMS, I suggest to add something in G to take in account number of time steps per year as an unavoidable constraint on the number of floating point operations. (Could also be in a next generation of those metrics)

We do not believe the temporal resolution should be included in $G$, for the following reason: it is chosen as a consequence of the spatial resolution and the accuracy one needs to achieve. For example, the timestep may be bounded by a CFL criterion once the spatial resolution is given, for one class of methods. For implicit methods where there is no CFL, it is still bounded by the level of accuracy needed. In any case, the timestep is never varied independently of the limits of physics and resolution purely for reasons of speed and performance, and is generally set to the largest value imposed by those limits. It therefore should not be separately accounted for in $G$.

9. Page 19: Reference for XIOS: is this paper from Joussaume et al. the appropriate reference for XIOS today?

Unfortunately, there does not yet exist a better reference to XIOS than Joussaume et al. (2012) and the URL to the source and documentation, already provided.

---

## Author Comment (AC3) · 26 Nov 2016

Response to gmd-2016-197-RC2: (note that some internal numbering from the review has been modified to create consistent numbering.

10. One can still debate whether other metrics should be considered as relevant to our field. While the paper correctly suggests the increasing importance of energy metrics in HPC – in this case the JPSY and AJPSY metrics – perhaps it should also consider other evolving metrics, e.g., FTTSE [Bekas, C. & Curioni, A. Comput Sci Res Dev (2010) 25: 187]. This comment is meant to encourage discussion and promote the evolution of a more complete set of metrics. As mentioned, the strength of the paper lies in the agreement of many institutions on the presented set.

   This is indeed a useful reference, and has been added to the text. As noted in the discussion of AJPSY (which is very similar to the FTTSE), we believe we will extend toward such metrics when the hardware routinely supports it. See revised discussion starting page 13, line 20.

11. Explicit mention of spectral models should be made on p. 5, line 1. Spectral models are still in use and tend to have very different scaling behaviour than FD-, FV- or FEbased models.

   Agreed, this is an oversight. Fixed, see page 5, line 6.

12. Point 6 on p. 10 (lines 4-6) is not necessary to mention here. It is repeated in section 3.2 and need not be emphasised.

   Agreed, we have reworded this section to make the general point instead of repeating what is below in Sec 3.2. See page 11, line 3.

13. Given the caveats mentioned about dimensionality on p. 11, lines 6-7, why even mention the nominal Cartesian representation of NX x NY x NZ on p. 11, line 1?

   Fair point: see paragraph of revised text beginning page 11, line 14.

14. There are several full sentences enclosed in parentheses (p. 12, lines 28-29; p. 14, lines 7-8; p. 15, lines 25-26; p. 19, lines 19-21). As a matter of style, it would be better to either (a) remove the parentheses if the authors feel these comments belong in the text, or (b) make them footnotes.

   Fixed; the parentheses have been replaced by a footnote in one instance (page 13, line 7); sentence eliminated altogether just below, as it repeats the footnote text (page 13, line 8); parentheses removed and sentence retained in two instances (page 14, line 19; page 16, line 14).

15. The paper is well-written and concise. Obviously great care has been taken in proofreading. Still a small number of mistakes have slipped through. GMD allows authors to write in the variety of English of their choice: We accept all standard varieties of English in order to retain the author's voice. However, the variety should be consistent within each article. When using Oxford spellings, please do so consistently. Whilst the vast majority of spellings are U.S. ("parallelization", "modeling", "characterize", "recognize", ...), the authors curiously use "centre" (8 occurrences, although 2 occurrences of "center"), and "summarised" (1 occurrence). Please use consistent American spellings, if this is your choice. Fixed;

 Additionally we found one instance of "modelling" which has now become "modeling", consistent with the rest of the text:

16. Various forms are used for "flops", such as "FLOP rates", "flop rates", and "flops". Please use one consistently.

5
We have now used "flops" consistently:

17. While there is no consensus in writing manuals whether there should be a comma after "i.e." and "e.g.", it is universally agreed that there should be a full stop after each letter. Please correct the instances of "i.e" and "e.g".

18. The four points at the end of p. 2 / beginning of p. 3 are part of the sentence on p. 2, lines 31-32. Please terminate these with semi-colons rather than full stops.

19. Same point for the 4 points in the sentence on p. 8, lines 15-22.

15

20. p. 7, line 1: "such as the recent AVEC report do take it " should be "... recent AVEC report, do take it".

21. p. 7, line 12: "... maybe run in ..." should be " ... may be run in ... ".

20
22. p. 15, line 1: "In this case. it may be ..." should be "In this case, it may be ...".

23. "S-mode" and "T-mode" are defined (quite appropriately!) on p. 9, line 13, but are used as "S mode" (p. 19, line 8) and "T mode" (p. 16, line 10; p. 19, lines 28-29). Please use consistently.

25
24. p. 21, line 1: "core for core, the new machine ..." should be "Core for core, the new machine ...".